# Textile-Based Flexible Capacitive Pressure Sensors: A Review

**DOI:** 10.3390/nano12091495

**Published:** 2022-04-28

**Authors:** Min Su, Pei Li, Xueqin Liu, Dapeng Wei, Jun Yang

**Affiliations:** 1School of Science, Chongqing University of Technology, Chongqing 400054, China; sumin@stu.cqut.edu.cn (M.S.); xqliu@cqut.edu.cn (X.L.); dpwei@cigit.ac.cn (D.W.); 2Chongqing Institute of Green and Intelligent Technology, Chinese Academy of Sciences, Chongqing 400714, China; 20160802049@cqu.edu.cn

**Keywords:** capacitive pressure sensor, textile, micro/nanostructure, flexibility, wearable electronics

## Abstract

Flexible capacitive pressure sensors have been widely used in electronic skin, human movement and health monitoring, and human–machine interactions. Recently, electronic textiles afford a valuable alternative to traditional capacitive pressure sensors due to their merits of flexibility, light weight, air permeability, low cost, and feasibility to fit various surfaces. The textile-based functional layers can serve as electrodes, dielectrics, and substrates, and various devices with semi-textile or all-textile structures have been well developed. This paper provides a comprehensive review of recent developments in textile-based flexible capacitive pressure sensors. The latest research progresses on textile devices with sandwich structures, yarn structures, and in-plane structures are introduced, and the influences of different device structures on performance are discussed. The applications of textile-based sensors in human wearable devices, robotic sensing, and human–machine interaction are then summarized. Finally, evolutionary trends, future directions, and challenges are highlighted.

## 1. Introduction

Over the past few decades, the world has seen the stunning progress of flexible pressure sensors in electronic skin [1,2,3,4,5,6], human movement [7,8,9,10], health monitoring [11,12,13,14,15,16,17], and human–machine interactions [18,19,20,21,22,23]. Compared with traditional rigid sensors, flexible sensors have the advantage of bendable and deformable substrates, which enables their application in wearable conformal devices. Flexible pressure sensors can be classified into many types, such as capacitive [24,25,26,27,28], resistive [29,30,31,32,33], piezoelectric [34,35,36,37,38], triboelectric [39,40,41,42,43,44,45,46,47,48], iontronic [49,50,51,52,53,54], and optical vacuum [55,56,57] sensors, based on their sensing mechanisms. Capacitive pressure sensors (CPS) have received extensive attention due to their stability, low power consumption, high response speed, simple structure, and low-cost scalable manufacturing process [58,59]. Commonly used flexible materials, such as polydimethylsiloxane (PDMS) and Ecoflex, are easy to deform; however, because of the limited compression range of these materials, the sensitivity of sensors fabricated from them is still insufficient. By introducing microstructures to modify the surface of the electrode and dielectric layer, the performance of a sensor can be significantly improved [59,60,61]. These microstructures increase the compressibility of the dielectric layer, causing it to deform under subtle pressure. In addition, voids in these structures introduce an air dielectric layer that changes the effective dielectric constant under pressure, resulting in larger capacitance changes. Photolithography molds are usually used in the preparation of traditional microstructures. This method can precisely control the shape of the microstructures, but the manufacturing method is expensive. Some scholars use natural plants as molds to prepare microstructures to save costs, but this method is not conducive to large-area preparation. Nanoimprint lithography is a high-resolution, low-cost method for fabricating large-area nanostructures, but the life of the mold needs to be further improved [62]. Therefore, introducing sensing materials with high-performance microstructures and low-cost manufacturing processes is critical to developing flexible pressure-capacitance sensors. Recently, textiles have attracted the attention of researchers in the application of wearable electronic products due to their unique advantages [63,64,65,66]. These textile-based materials have excellent flexibility to fit the user’s body shape and have good air permeability and abrasion resistance for long-term wearable applications. Meanwhile, a flexible pressure sensor’s unique three-dimensional porous structure can introduce or increase the size of air gaps and voids to improve the device’s performance. In addition, textiles are inexpensive and easy to fabricate in a large area, making them a new material with great potential in wearable pressure sensors.

Although the research progress of textile-based flexible pressure sensors is discussed in some articles [67,68,69], a comprehensive overview of textile-based flexible capacitive pressure sensors has not been fully reported in the literature. This work aims to fill this knowledge gap, hoping to provide guiding ideas for improving and innovating flexible textile-based capacitive pressure sensors. The four key contributions of this paper are presented as follows. First, starting from the working principle of capacitive pressure sensors, the existing textile-based capacitive pressure sensors are divided into five forms according to the different functions of textiles in capacitive pressure sensors. Second, according to the form of the sensor and the selected material, the preparation methods of the textile-based capacitive pressure sensor can be summarized as weaving technology, fabric substrate modification, and electrospinning technology. Third, textile-based capacitive pressure sensors can be divided into sandwich, yarn, and in-plane devices. Last, the potential applications of textile-based capacitive pressure sensors are summarized from the aspects of human wearable devices, robot sensing, and human–machine interaction.

This paper reviews the latest developments in capacitive pressure sensors based on textile structures. Section 2 describes the working principle of capacitive pressure sensors and the different forms of textile-based functional layers. In Section 3, materials and fabrication methods are discussed by considering the requirements for flexible electrodes and dielectric layers. Section 4 describes three devices types based on functional textile layers. Section 5 presents practical applications of textile-based capacitive pressure sensors. Finally, in Section 6, conclusions and outlooks are also put forward. Figure 1 shows the framework of the textile-based capacitive pressure sensor.

## 2. Flexible Capacitive Pressure Sensors

### 2.1. Working Principle

Table 1 compares the advantages and disadvantages of different sensors, including capacitive, resistive, piezoelectric, triboelectric, and iontronic sensors. Capacitive pressure sensors are mainly composed of electrodes and dielectric layers. Generally, capacitive pressure sensors use a ‘sandwich’ structure, in which the dielectric layer is sandwiched between the upper and lower electrodes. The capacitance value *C* is calculated by the formula [70]:(1)$$C=\frac{\varepsilon_{0}\varepsilon_{r}A}{d}$$
where *ε*_0_ is the vacuum dielectric constant, *ε_r_* is the relative dielectric constant of the dielectric layer, *A* is the effective area of the upper and lower electrodes, and *d* is the distance between the two electrode plates. By applying pressure, *ε_r_*, *A*, *d*, and the capacitance of the sensor change due to the elastic deformation of the soft material in the flexible capacitive pressure sensor. The formula shows that the capacitive pressure sensor can be divided into plate spacing, area, and dielectric change types [71]. The easiest way is to change the distance between the upper and lower plates of the sensor. Under pressure, the distance between the two polar plates decreases, and the capacitance of the sensor increases accordingly. However, a capacitive pressure sensor with a planar structure has low sensitivity due to the limited compression. There are three effective ways to improve the sensitivity of a sensor [72]: (1) constructing microstructures on the surface of dielectrics or electrodes [73]; (2) adding conductive fillers to the polymer elastomer to form a composite dielectric [74,75]; and (3) introducing micropores in the dielectric layer [76]. Building microstructures on a flexible material will help the sensor produce more significant deformation under a smaller pressure, thereby improving the sensor’s performance. In addition, since the dielectric constant of air is lower than that of the polymer, the discharge of air under an external force increases the overall dielectric constant, thereby improving the sensitivity of the capacitive pressure sensor [71]. Various microstructures are applied to electrodes or dielectric layers, including micropyramids, micropillars, microspheres, microdomes, micropores, and other micropatterns reversely shaped by the micro/nanostructures of natural plants [28,77,78,79,80,81,82,83,84,85,86,87,88,89,90,91,92]. Besides, fabrics and nanofiber membranes are also widely used due to their excellent properties, such as flexibility, air permeability, and compressibility [93,94,95,96]. Collectively, the applications of various microstructures are shown in Figure 2.

### 2.2. Functional Textile Layers

According to the CPS device structure composed of the electrode and dielectric layers, textile-based flexible capacitive pressure sensors are divided into five forms: (1) textile-structured electrodes, (2) textile-structured dielectric layers, (3) all-textile structures, (4) yarn structures, and (5) in-plane structures. These structures are shown in Figure 3. The first three forms all belong to the ‘sandwich’ structure. Under pressure, the greater the changes in *ε_r_*, *A*, and *d*, the higher the sensitivity. However, the conductive fabric electrode’s original excellent performance (such as flexibility and air permeability) is usually sacrificed after modification, and its stability is reduced. In addition, the compressibility of the fabric used for the dielectric layer is limited, resulting in the low sensitivity of the sensor. To improve the performance of textile-structured pressure sensors, an increasing number of scholars are focusing on nanofiber membranes prepared by electrospinning [97]. The thickness, porosity, and dielectric constant of the nanofiber membrane dielectric layer can be effectively controlled, so the sensor’s performance can be significantly improved. However, the first three forms of capacitive pressure sensors usually require a large surface, which is not conducive to integrating sensors in clothing. Therefore, some scholars have prepared sensors based on yarn structures and stitched the sensors on wearable clothing through the knitting method [98,99]. Capacitive pressure sensors based on yarn are better and more comfortable to wear and have a bright blueprint in smart clothing; however, their performance needs further improvement. Table 2 summarizes the structure and performance of different forms of textile-based capacitive pressure sensors.

## 3. Materials and Fabrication Methods for Textile Layers

A flexible capacitive pressure sensor has two main parts: flexible electrodes and dielectric layers. Electrodes and dielectric layers based on textile structures need to be improved in materials and preparation methods.

### 3.1. Materials Classification

Common textiles are usually composed of insulating materials, such as cotton and polyester fibers. These insulating materials need to be converted into conductive fibers through the coating, magnetron sputtering, screen printing, and other methods for sensor structures. At present, metal materials (Ag, Cu, Ni, etc. [114,115]), carbon-based materials (carbon nanotubes, graphene, etc. [116,117]), and some conductive polymers (poly(3,4-ethylene dioxythiophene): polystyrene sulfonic acid (PEDOT: PSS), polypyridine (PPy), polyaniline (PAN), etc. [118,119,120,121,122]) are commonly used in electrodes with flexible textile structures. These materials have advantages and disadvantages. Metals are the most commonly used conductive material due to their excellent ductility and conductivity. However, metal materials are rigid, limiting their application in wearable electronic devices. Carbon-based materials have a large active area, high electrical conductivity, and good electrochemical activity, but their processing is complicated, and the cost is high. Conductive polymers have attracted the attention of researchers because of their excellent flexibility, but their stability, conductivity, and processing need to be further improved.

The dielectric layer materials of flexible capacitive pressure sensors based on textile structures can be roughly divided into textile and nanofiber dielectric layers. The former type mainly includes the warp-knitted spacer type and woven structure type. Spacer fabric is thicker, deforms more under pressure, and has a relatively high detection range. However, the sensitivity of sensors based on textile dielectric layers is low, and the pore structure and dielectric properties of dielectric layers are uncontrollable. Nanofiber dielectric layers are nanofiber membranes with textile structures prepared by electrospinning techniques using inorganic materials or polymer solutions. Commonly used polymer materials are polyurethane (PU), polyvinyl alcohol (PVA), polyimide (PI), polyvinylidene fluoride (PVDF), and PVDF copolymers [102,105,123,124,125]. However, nanofiber membranes made of inorganic materials, such as TiO_2_ [126], are rarely used in capacitive pressure sensors. In addition, the application of composite materials can improve the performance of electrospun nanofiber membrane dielectric layers to a certain extent [94,127,128,129]. For example, Yang et al. [129] verified that adding a small number of carbon nanotubes near the percolation threshold of polymer solutions can significantly increase the dielectric constant. Therefore, polymer nanofibers prepared via electrospinning technology have the advantages of high flexibility, controllable structures, and good dielectric properties.

### 3.2. Fabrication Methods

Researchers continue to improve and innovate flexible textile pressure sensors with the continuous development of new sensing materials and manufacturing processes. The selected preparation methods are different depending on the sensor morphology and material. There are currently three main methods: weaving technology, fabric substrate modification, and electrospinning technology.

#### 3.2.1. Weaving Technology

Metal-covered or other conductive yarn can be attached to a textile substrate by embroidery or weaving. Seiichi Takamatsu et al. [118] developed a meter-scale large-area capacitive fabric pressure sensor used as a floor sensor to monitor the position of the human body (Figure 4a). In the fabric pressure sensor, two fabrics woven with strip electrodes of conductive polymer-coated fibers were stacked vertically, and the capacitance difference between the top and bottom strip electrodes was measured when pressure was applied. Simge Uzun et al. [99] coated cellulose yarns with Ti_3_C_2_T_x_ MXenes to produce highly conductive and electroactive yarns that can be woven into textiles using industrial knitting machines (Figure 4b). However, it is expensive to prepare ordinary nonconductive yarns into conductive yarns through special treatment and then integrate them with the clothing through weaving. Therefore, Talha Agcayazi et al. [129] fabricated capacitive sensor networks from conductive sewing threads (silver-coated polyamide (silver) and stainless steel (SS)) by conventional sewing processes. In this configuration, the fabric is the dielectric, and the conductive yarn acts both as an inductive capacitive ‘electrode’ and as a line to connect to an external front-end circuit (Figure 4c). A flexible capacitive pressure sensor prepared by the weaving method retains the original textile structure of the clothing and can fit the three-dimensional curved surface of the human body. Its integration into wearable electronic products is also more flexible than the sandwich structure, which has attracted the attention of many scholars. However, there are still many problems, such as the design of yarn structure, optimization of sensor performance, slip under pressure, etc. In addition, the design of the large-area woven array readout circuit remains to be further studied.

#### 3.2.2. Fabric Substrate Modification

For preparing traditional textile-based pressure sensors, coating methods (magnetron sputtering, screen printing, chemical deposition, etc.) are usually used to deposit various advanced conductive nanomaterials on textiles, which can fabricate conductive electrodes in capacitive pressure sensors. Li et al. [10] used nickelized polyester yarn to make a conductive tape, and then electroplated a layer of copper onto the tape to improve its conductivity, as shown in Figure 5a. Wu et al. [101] used magnetron sputtering to prepare a silver-plated conductive fabric with a twill structure, and its conductivity was as high as 0.268 Ω·cm (specific resistance). The conductive fabric was combined with a medium layer consisting of the elastomer Ecoflex to prepare a flexible wearable pressure sensor. Chen et al. [131] used cotton fabric as the essential component and developed a topological modification method with 3-genus and 5-genus structures. Topological genus structures can form cage-like metal seeds on the substrate surface. The conductive material was uniformly wrapped around the cotton fibers, forming a highly conductive interconnected network (Figure 5b). Golabzaei et al. [132] sprayed PET fabric with graphite solution and used screen printing to add PEDOT: PSS coating for improving the fabric’s conductivity. Compared with a sample coated with graphite only (3 kΩ), the electrode resistance of the coating after adding PEDOT: PSS could reach 300 Ω. Of course, the method of preparing conductive fibers is not limited to the coating. Carbonized cotton fabric (CCF), converted from cotton fabric by simple pyrolysis, has good flexibility and conductivity, which is an ideal material for flexible pressure sensors. More importantly, the carbonization process is scalable, low-cost, and eco-friendly. Ko et al. [103] combined a carbonized cotton fabric (CCF) electrode with a porous Ecoflex dielectric layer, as shown in Figure 5c. The sensor exhibited high sensitivity, which is attributed to the enhanced deformability of the medium and the roughness of the electrode textile structure. Carbonization methods are used for fewer applications regarding flexible capacitive pressure sensors and require further material optimization and process design to ensure the repeatability and integrity of their performance. The coating method is relatively simple in the modification process of fabric substrates, and it is easy to control the reaction conditions. However, the adhesion problem between the fabric substrate and conductive material affects the conductivity of the fabric and destroys the original textile structure of the fabric. Therefore, to improve the performance of flexible fabric sensors, it is necessary to find new processes or preparation methods to modify or optimize the structure of conductive fabrics.

#### 3.2.3. Electrospinning Technology

Electrospinning is a direct, efficient, and scalable technology for preparing nanofiber membranes with textile structures [133]. The nanofiber structure prepared by this technology has a low Young’s modulus and a high surface-to-volume ratio and porosity (approximately 70%). Due to the low Young’s modulus of the fiber membrane, a large compression deformation can be achieved under slight external pressure. Therefore, in capacitive pressure sensors, electrospinning has attracted increasing attention. Wang et al. [109] used screen printing to coat carbon nanotubes (CNTs) on a TPU nanofiber membrane as a flat electrode, as shown in Figure 6a. The process is simple and maintains the flexibility and air permeability of the electrode. In the deposition or printing process, it is necessary to pay attention to the uniformity of the coating and the adhesion between the different coatings and the textile substrate. The performance of conductive fibers can be improved by special treatment of conductive materials or by selecting different process combinations. Chen et al. [102] prepared a flexible electrode by electrospinning a palladium ion (Pd^2+^)/polyacrylonitrile (PAN) solution and then electroless plating the mixed nanofiber membrane, as shown in Figure 6b. The field emission scanning electron microscopy (FESEM) image shows that the obtained conductive nanofiber membrane is covered by a coral-like silver layer and porous, which is beneficial for achieving excellent conductivity. Wang et al. [115] prepared a transparent copper/nickel nanonetwork based on electrospinning and chemical deposition, ensuring the sensor’s high bending and cycle stability. The performance of a conductive fiber electrode prepared by electrospinning can be further optimized by controlling the material selected, the thickness, and the porosity of the fiber membrane.

The dielectric layer of flexible capacitive pressure sensors can be divided into the porous elastomer dielectric layer, the polymer film dielectric layer, and the textile-structured dielectric layer. A dielectric layer with a textile structure has good flexibility and air permeability. Most textile-structured dielectric layers are polymer nanofiber membranes, ionic fiber membranes, or composite fiber membranes prepared with electrospinning. For example, Li et al. [132] used electrospinning to prepare a dual-structured polyurethane nanofiber membrane (TPU NM) dielectric layer in a flexible capacitive pressure sensor, as shown in Figure 6c. Due to the prolific air in the dielectric layer, the designed sensor demonstrated outstanding sensing performance with high sensitivity (0.28 kPa^−1^) in the low-pressure region (0–2 kPa), fast response/relaxation (65/78 ms), and high-grade durability (1000 cycles). Sharma et al. [89] used electrospinning to prepare MXene (Ti_3_C_2_T_x_)/ PVDF-trifluoroethylene (PVDF-TrFE) composite nanofiber membranes, which were used in the dielectric layer of capacitive pressure sensors for ultralow pressure measurement (Figure 6d). The sensor had a high sensitivity of 0.51 kPa^−1^, and the minimum detection limit was 1.5 Pa. The fiber structure is an effective structure to reduce the compressive modulus of materials further and improve the sensitivity of devices. Moreover, the synergy of the two different materials in a composite fiber membrane can further improve the performance of sensors.

In general, these studies suggest that in the preparation of flexible textile capacitive pressure sensors, knitting retains the original textile structure of the clothing, making the fabricated sensor more suitable for three-dimensional curved surfaces and more flexibly integrated into wearable electronic products. However, the structural design and performance optimization of the fibers or yarns used for weaving needs to be further studied, and the problem of sensor slippage under pressure has yet to be resolved. The method of fabric substrate modification is relatively simple, and the selection of conductive materials and attachment methods is relatively flexible. However, the uneven coating and easy shedding between the fabric substrate and the conductive material damages the fabric’s original textile structure and affects its conductivity. Electrospinning can control the thickness and porosity of the fiber membrane through parameter settings. The fiber membrane and the coating material can be better combined through material selection, and the sensor performance and use performance can be further improved.

## 4. Textile-Based Flexible Capacitive Pressure Sensors

From textiles with a simple structure and low compression performance to applications of nanofiber membranes prepared by electrospinning, the device types of textile-based capacitive pressure sensors are mainly divided into the sandwiched devices, yarn devices, and in-plane devices. The sandwich devices can be subdivided into the semi-textile structure and all-textile structure. The semi-textile structure means that one of the counterparts of the sensor, either the electrodes or the dielectric layer, is a textile.

### 4.1. Sandwich Devices

Atalay et al. [100] designed a capacitive pressure sensor composed of two soft conductive fabrics (knitted fabric and woven fabric) and two microporous dielectric layers (sugar particles and salt crystals). They evaluated the effects of the components on the sensor’s overall performance (Figure 7a). The study found that the conductive knitted electrodes and higher dielectric porosity (attributed to the large sugar particles) resulted in higher sensitivity (121 × 10^−4^ kPa^−1^). The higher the porosity of the dielectric layer, the looser the electrode structure. Therefore, the deformation of the dielectric layer under pressure is more significant, which is beneficial to the sensor’s performance. The simplest way to optimize textile-structured electrodes is to use electrospinning to prepare nanofiber membranes as substrates and convert them into conductive fibers by sputtering and other methods. This approach optimizes the structure and thickness of the electrodes and ensures flexibility and breathability.

Other optimization methods can build better-structured textile dielectric layers to improve sensor performance. Some researchers have prepared micropatterned nanofiber membranes through process design. Jin et al. [135] used an electrospinning process to prepare dielectric membranes composed of insulating microbeads within polyvinylidene fluoride (PVDF) nanofibers (Figure 7b). The presence of microbeads increases the porosity, increasing the sensor’s sensitivity (1.12 kPa^−1^ in the range of 0 to 1 kPa). Yu et al. [110] prepared a high-sensitivity, ultrathin, all-fabric capacitive pressure sensor based on a gas-permeable network with a micropatterned nanofiber dielectric layer (Figure 7c). The sensor exhibited high sensitivity (8.31 kPa^−1^ at 1 kPa), a low detection limit (0.5 Pa), a wide detection range (0.5 Pa–80 kPa), good robustness (10,000 cycles), and exceptional air permeability. However, in previous research on capacitive devices with all-textile structures, the performance of capacitive pressure sensors with three-layer fabrics is often low. For example, Vu et al. [108] used a PET yarn layer as a dielectric layer and highly stretchable printed Ag/SWCNT fabric as an electrode, and the prepared sensor had a sensitivity of up to 0.042 kPa^−1^. Nanofibers prepared by electrospinning are beneficial for improving the performance of all-textile capacitive pressure sensors, and nanofiber membranes with micropatterns or unique structures can further improve the performance of sensors.

### 4.2. Yarn Devices

Clothing is made from countless fibers or yarns through weaving. Some researchers have created yarn-structured sensors to make real-time monitoring of human health and movement more comfortable and convenient. Ashok Chhetry et al. [107] prepared a flexible, high-sensitivity capacitive pressure sensor by coating microporous PDMS elastomer dielectric on conductive fibers, as shown in Figure 8a. The sensor consisted of a microporous dielectric with a sensitivity of 0.278 kPa^−1^ (<2 kPa), a response time in the millisecond range (~340 ms), and a dynamic range from 0–50 kPa. You et al. [136] constructed a wearable electronic fabric based on a stretchable capacitive sensor array woven by electrospun nanofiber-coated yarns, as shown in Figure 8b. The fabric was electrospun to coat graphene oxide (GO)-doped polyurethane (PU) nanofibers on the surface of nickel-plated cotton yarn. Then, the nickel-plated cotton yarn coated with nanofibers was wound around the elastic thread. The sensor unit had high sensitivity (1.59 N^−1^, <0.3 N), a wide sensing range (0–5 N), a low detection limit (0.001 N), and a short response time (<50 ms). Zhang et al. [113] proposed a stitch-woven structure sensor with silver fibers and cotton fibers as the electrode and the dielectric layer of the capacitive sensor. Compared with the traditional sandwiched structure, a sensor with this structure can be integrated into any fabric position without being restricted by the shape, solving the problem of pressure sensing in irregular areas of fabric. However, compared with ordinary knitted yarns, conductive yarns have greater rigidity, which is not favorable for torsional deformation. Moreover, the sensing part is prone to slippage between yarn strands under pressure, which affects the stability and accuracy of the sensors. Therefore, capacitive pressure sensors based on the yarn structure need to be optimized for their materials and structures to reduce their stiffness and address slippage problems.

### 4.3. In-Plane Devices

Among the research progress on textile-based capacitive pressure sensors, the sandwich structure and yarn structure are widely studied, and the in-plane structure is slowly emerging. Ozgur Atalay [137] introduced a sensing structure that combines conductive woven fabric and a silicone elastomer. The stretchable conductive woven fabric acts as an interdigital electrode, creating a secure conductive network. The silicone elastomer fills the area between the electrodes, forming a dielectric layer and encapsulating the structure (Figure 9a). MD Abdullah al Rumon [138] used commercial stainless steel thread and a low-cost sewing machine to prepare a low-cost, expandable tactile sensor that senses the tactile and conjugate pressure on fabric through changes in capacitance (Figure 9b). When a finger contacts the interdigital electrodes, the dielectric constant of the sensor increases, thereby increasing the capacitance. Compared to the parallel plate capacitor structure, the interdigitated electrode structure creates a more functional area due to having more electrode arrangements. Researchers have shown that the fringing capacitance between interdigital electrodes depends on the dielectric layer’s parametric properties and the electrodes’ size [139]. Therefore, device performance improves by optimizing the electrodes and the dielectric layer for interdigital electrode fabric capacitive pressure sensors.

Compared to the thicker planar structure, the looser textile structure improves the sensor’s performance. The performance of a semi textile structure sensor that only uses fabric as the electrodes or the dielectric layer is poor. The main reason for this may be the fabric electrode’s poor conductivity and the dielectric layer’s poor compressibility. This problem is better addressed by using electrospun nanofibrous membranes instead of traditional fabrics. The researchers improved the conductivity of the conductive fibers by choosing electrospinning materials and coating materials. The structure and thickness of the fiber membrane can be optimized by changing the electrospinning parameters so that the dielectric layer deforms significantly under slight pressure. In addition, the utilization of micropatterned or specially structured nanofiber membranes can further enhance the performance of all-textile capacitive pressure sensors. A capacitive sensor with a sandwich structure has the advantages of good flexibility, high sensitivity, and short response time. However, it requires a larger flat surface and a relatively large thickness, and it still lacks wearability compared to yarn. A sensor based on the yarn structure is easier to weave into clothing and can better fit three-dimensional surfaces. It is difficult for the in-plane devices to prepare stable and high-density interdigitated electrodes on fabric substrates because fabric easily deforms and stretches.

## 5. Applications

Capacitive pressure sensors based on the textile structure can be integrated into various products daily, such as gloves, socks, seats, mattresses, etc., and are primarily used in wearable devices and robot tactile interactions.

### 5.1. Wearable Devices

Physiological signals (pulse, blood pressure, respiration rate, etc.) play an essential role in health monitoring. Textile-based capacitive pressure sensors can measure weak pressure signals and have the advantages of being breathable and comfortable and fitting the human body well. Yang et al. [106] showed a skin-type pressure sensor with high sensitivity, good flexibility, fast response, good air permeability, and light weight, making it suitable for low-cost, large-area production (Figure 10a). This skin-type pressure sensor can monitor physiological signals such as human respiration and heart rate. The air permeability of its all-textile structure and the simple preparation process provide a promising strategy for designing air-permeable electronic skins. Wu et al. [107] successfully produced an all-textile pressure sensor and wireless battery-less monitoring system to detect human movement in real-time, in which a 3D permeable fabric was sandwiched between two highly conductive fabric electrodes as a dielectric layer (Figure 10b). Posture monitoring is also vital in medical diagnosis and safety protection. Accurately monitoring the pressure on various body parts (including the soles of the feet, legs, back, chest, and neck) helps testers correct their postures in time to prevent the formation of foot ulcers, bedsores, and other diseases. Vu et al. [108] proposed a multipurpose capacitive textile pressure sensor for wearable electronic applications embedded in intelligent socks that can be used for walking gait analysis in daily life activities (Figure 10c). Masihi et al. [140] combined a porous PDMS dielectric layer and two conductive fabric electrodes to form a capacitive pressure sensor, and used a sensor array to measure and map the pressure on a player’s head when wearing a helmet (Figure 10d). The pressure distribution diagram helps the user observe and adjust the proper position.

### 5.2. Robotic Sensing

With the development of wearable electronic products, the application of pressure sensors in tactile sensing has attracted the interest of many researchers. Ozgur Atalay et al. [100] demonstrated the application of a soft pressure sensor to detect grasping force via the integration of the sensor into a textile glove (Figure 11a). They noted that such sensors could also be used in soft wearable robotics. Therefore, textile-based capacitive pressure sensors have excellent prospects in robotic sensing. Ahmed Elsayes et al. [141] fabricated capacitive tactile sensors by sandwiching a microstructured dielectric elastomer layer between two conductive fabric electrodes. These sensors are integrated into an anthropomorphic manipulator fabricated using rapid prototyping techniques. Lin et al. [96] prepared a new type of capacitive pressure sensor using a double-layer dielectric structure composed of electrospun fibers and microcylinder arrays and used it to monitor robotic arm grasping objects in real-time (Figure 11b).

### 5.3. Human–Machine Interaction

Textile electronic devices may provide a suitable platform for human–machine interaction applications due to their superior performance and unique immersive properties, such as light weight, flexibility, and comfort. Lee et al. [111] prepared a fabric capacitive pressure sensor based on highly conductive fibers coated with an insulating rubber material. The sensor was used to control unmanned aerial vehicle (UAV) quadrotors based on its excellent performance. The four textile-based pressure sensors on the glove correspond to the four different motions of the quadcopter (index finger: right flight; middle finger: forward flight; ring finger: left flight; and little finger: backward flight). In the same way, the four pressure sensors were stitched on the forearm of clothes to control a wired hexapod walking robot. The four pressure sensors controlled the robot’s different motion commands: back (channel 1), forward (channel 2), counterclockwise rotation (channel 3), and clockwise rotation (channel 4), as shown in Figure 12a. Zhao et al. [104] integrated ten independent capacitive tactile sensors on gloves. The collected capacitive signals were processed and then transmitted to a mobile phone application via Bluetooth to perform a virtual piano performance, as shown in Figure 12b. Talha Agcayazi et al. [130] demonstrated the application of an all-textile capacitive pressure sensor in a smart glove for a drone’s remote control. A 3 × 3 pressure sensor array was sewn on the back of a PET glove to serve as the remote control for the quadrotor drone. Selecting four sensing points means that the drone moves in four directions (top sensing point: fly up; bottom sensing point: fly down; right sensing point: fly right; and left sensing point: fly left), as shown in Figure 12c.

## 6. Conclusions and Outlooks

Textile-based functional layers can introduce both porous airgaps and micro/nanostructures to enhance the performance of flexible capacitive pressure sensors. In addition, textile-based capacitive pressure sensors exhibit excellent flexibility, breathability, and comfort, making them easy to integrate with clothing, and have great potential in wearable electronics. This review presents the current research and progresses in textile-based capacitive pressure sensors. According to the sensing principle and device structures, the functional textile layer can be divided into five forms: textile-structured electrodes, textile-structured dielectric layers, all-textile structures, yarn structures, and interdigital electrode structures. Then, materials and fabrication methods for functional textile layers are discussed by considering the requirements for flexible electrodes and dielectric layers, including weaving technology, fabric substrate modification, and electrospinning technology. Three types of devices with the sandwich, yarn, and in-plane structures are discussed for textile-based sensors. Finally, the textile-based capacitive pressure sensor applications in human wearable devices, robot sensing, and human–machine interactions are demonstrated, indicating its great application potential in wearable intelligent electronic devices.

Textile-based flexible capacitive pressure sensors need to achieve excellent sensing performance (sensitivity, response time, repeat stability, etc.) and ensure that their flexibility, air permeability, durability, and other performance meet the requirements of wearable electronic devices. However, there are still many limitations in the fabrication process and application of the existing devices:The fabrication method needs to be further optimized. Problems such as easy peeling and unevenness of conductive materials on the surface of the textile-based electrode will affect their conductivity and durability. Therefore, there is a need to develop an efficient method for fabricating textile-based electrodes. In addition, the dielectric properties, compressibility, and stretchability of textile-based dielectric layers also need to be further improved.Although textile-based capacitive pressure sensors have many applications in smart textile clothing, current smart clothing is not washable. Furthermore, the low-cost and efficient fabrication of large-area textile-based capacitive pressure sensor arrays has not been reported yet. Other aspects, such as reducing crosstalk between capacitive signals, multimodal detection, etc., still need further research.

In the future, the way to improve the performance of sensors lies in broadening the exploration of new materials and developing new processes. Moreover, multifunctional integration studies of textile-based sensors can be conducted for the development of comfortable, low-cost, multifunctional smart textile garments.

## Figures and Tables

**Figure 1 nanomaterials-12-01495-f001:**
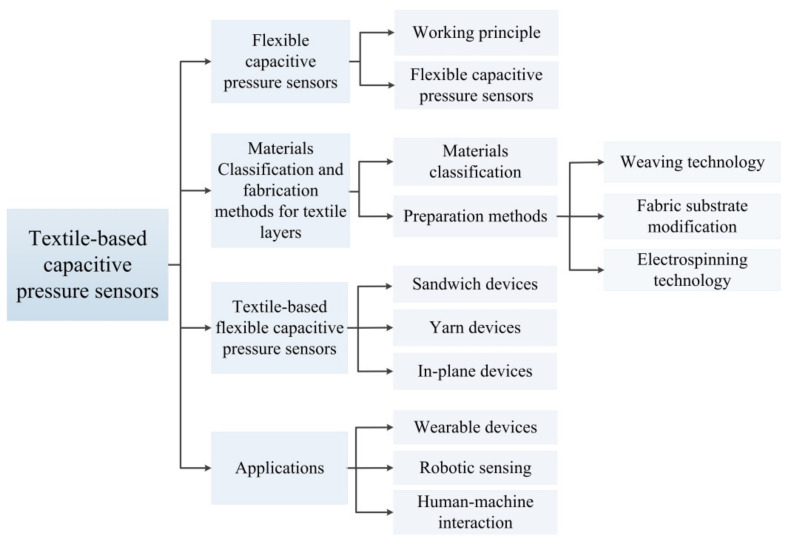
The framework of textile-based capacitive pressure sensors.

**Figure 2 nanomaterials-12-01495-f002:**
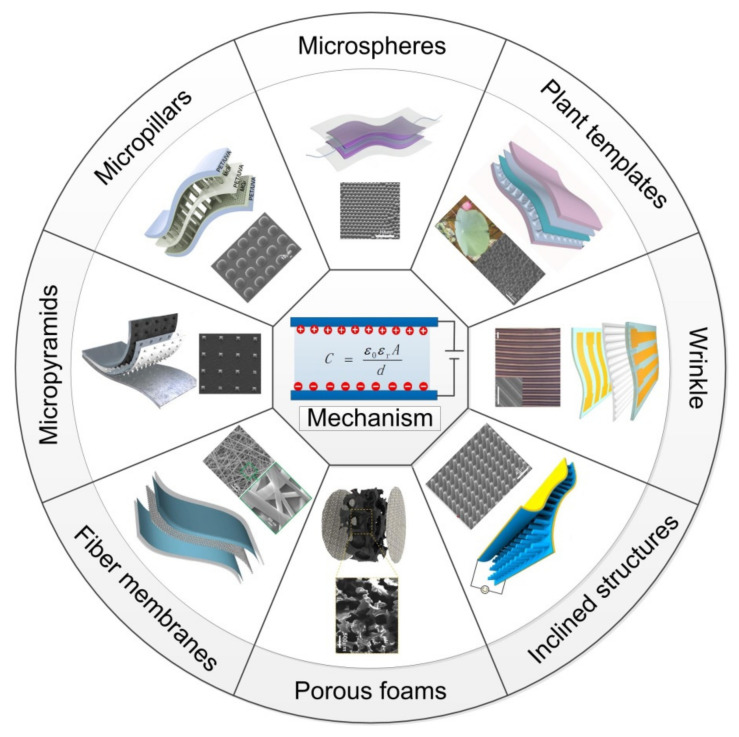
Capacitive pressure sensors based on different microstructures. (Micropyramids: Reprinted with permission from Ref. [81]. Copyright 2021 Elsevier. Micropillars: Reprinted with permission from Ref. [80]. Copyright 2019 American Chemical Society. Microspheres: Reprinted with permission from Ref. [82]. Copyright 2020 Elsevier. Plant templates: Reprinted with permission from Ref. [79]. Copyright 2018 Wiley-VCH. Wrinkle: Reprinted with permission from Ref. [90]. Copyright 2019 American Chemical Society. Inclined structures: Reprinted with permission from Ref. [89]. Copyright 2019 American Chemical Society. Porous foams: Reprinted with permission from Ref. [92]. Copyright 2020 American Chemical Society. Fiber membranes: Reprinted with permission from Ref. [94]. Copyright 2020 American Chemical Society).

**Figure 3 nanomaterials-12-01495-f003:**
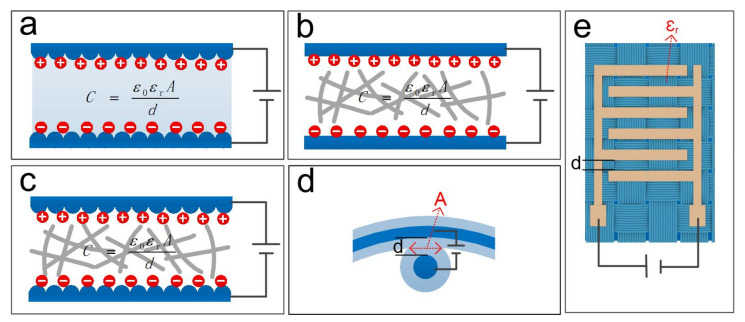
Capacitive pressure sensors based on different functional textile layers: (**a**) Textile-structured electrodes; (**b**) Textile-structured dielectric layers; (**c**) All-textile structures; (**d**) Yarn structures; (**e**) In-plane structures.

**Figure 4 nanomaterials-12-01495-f004:**
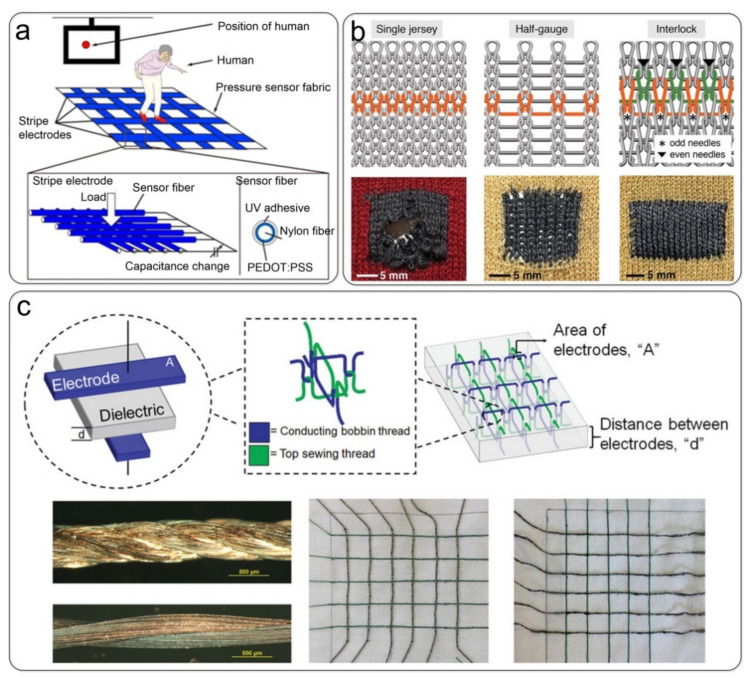
Fabrication of a capacitive pressure sensor by weaving technology. (**a**) Vertically stacked strip fabric electrodes of conductive polymer-coated fibers. Reprinted with permission from Ref. [118]. Copyright 2015 Springer Nature. (**b**) Different knitting patterns of multifunctional MXene-coated cellulose yarns. Reprinted with permission from Ref. [99]. Copyright 2019 Wiley-VCH. (**c**) Seam-line sensor network produced with silver threads. Reproduced with permission Ref. [130]. Copyright 2020 Wiley-VCH.

**Figure 5 nanomaterials-12-01495-f005:**
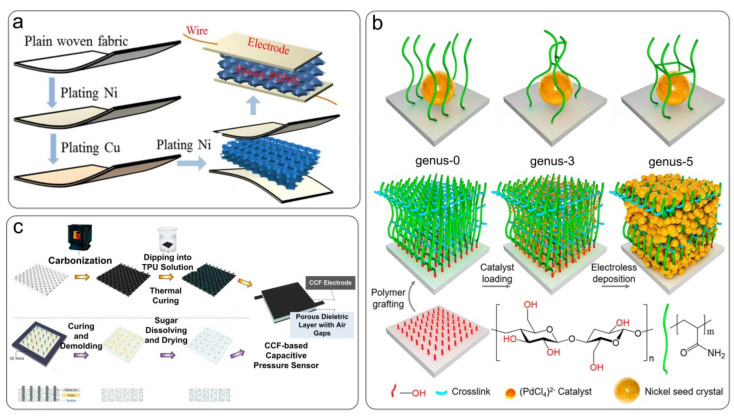
Fabric substrate is modified to prepare a capacitive pressure sensor. (**a**) Conductive fabric electrodes electroplated with nickel and copper. Reproduced with permission from Ref. [10]. Copyright 2020 Elsevier. (**b**) Preparation of cotton fibers with a uniform coating of conductive materials using a topology modification method. Reproduced with permission from Ref. [131]. Copyright 2020 American Chemical Society. (**c**) Preparation of a fabric capacitance sensor by combining a carbonized cotton fabric electrode with an Ecoflex dielectric layer. Reproduced with permission from Ref. [103]. Copyright 2021 MDPI.

**Figure 6 nanomaterials-12-01495-f006:**
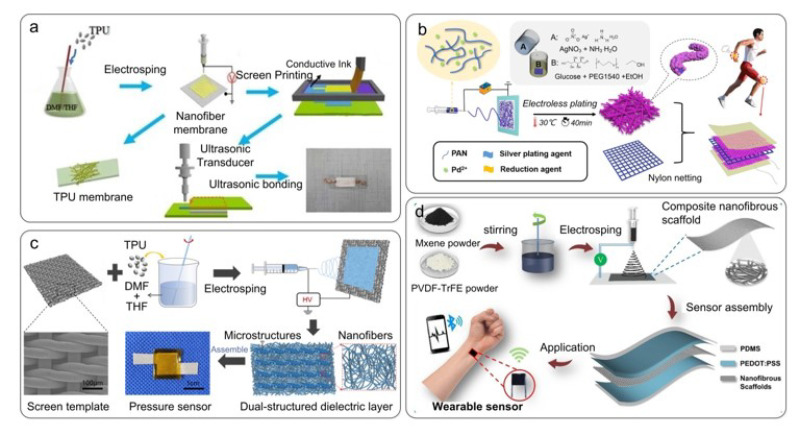
Capacitive pressure sensor prepared by electrospinning technology. (**a**) Screen printing technology coated CNTs on Electrospun TPU nanofiber membranes as planar electrodes. Reproduced with permission from Ref. [109]. Copyright 2021 MDPI. (**b**) Electrospinning with Pd^2+^/PAN solution and electroless plating of a mixed nanofiber membrane to prepare a flexible electrode. Reproduced with permission from Ref. [102]. Copyright 2020 Elsevier. (**c**) Preparation of a dual-structure polyurethane nanofiber membrane by electrospinning. Reproduced with permission from Ref. [134]. Copyright 2022 American Chemical Society. (**d**) Preparation of an MXene/PVDF-TrFE composite nanofiber membrane and its use as the dielectric layer of a capacitive pressure sensor. Reproduced with permission from Ref. [91]. Copyright 2020 American Chemical Society.

**Figure 7 nanomaterials-12-01495-f007:**
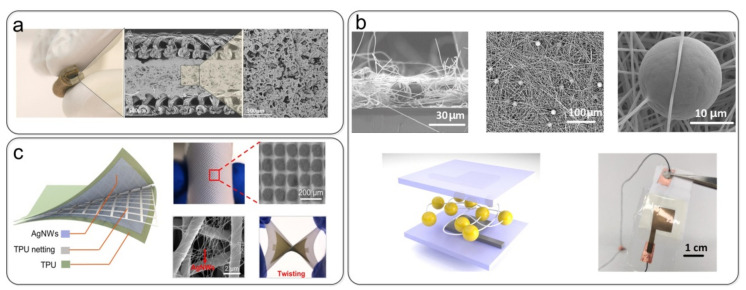
Capacitive pressure sensor based on the sandwich devices. (**a**) Combining soft conductive fabrics (knitted and woven fabrics) and microporous dielectric layers (sugar particles and salt crystals) to prepare capacitive pressure sensors. Reproduced with permission from Ref. [100]. Copyright 2017 Wiley-VCH. (**b**) Electrospinning to prepare dielectric membranes composed of insulating microbeads within PVDF nanofibers. Reproduced with permission from Ref. [135]. Copyright 2020 American Chemical Society. (**c**) All-fabric capacitive pressure sensor based on a micropattern nanofiber dielectric layer. Reproduced with permission from Ref. [110]. Copyright 2021 American Chemical Society.

**Figure 8 nanomaterials-12-01495-f008:**
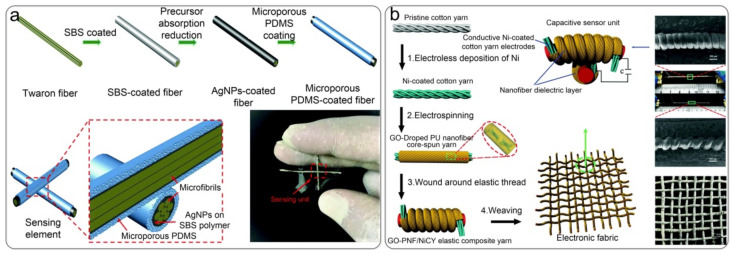
Capacitive pressure sensor based on the yarn devices. (**a**) Coating microporous PDMS elastomer dielectric on conductive fibers to prepare a capacitive pressure sensor. Reproduced with permission from Ref. [112]. Copyright 2017 Royal Society of Chemistry. (**b**) Retractable capacitance sensor array weaved by electrospun nanofiber-coated yarn. Reproduced with permission from Ref. [136]. Copyright 2017 Royal Society of Chemistry.

**Figure 9 nanomaterials-12-01495-f009:**
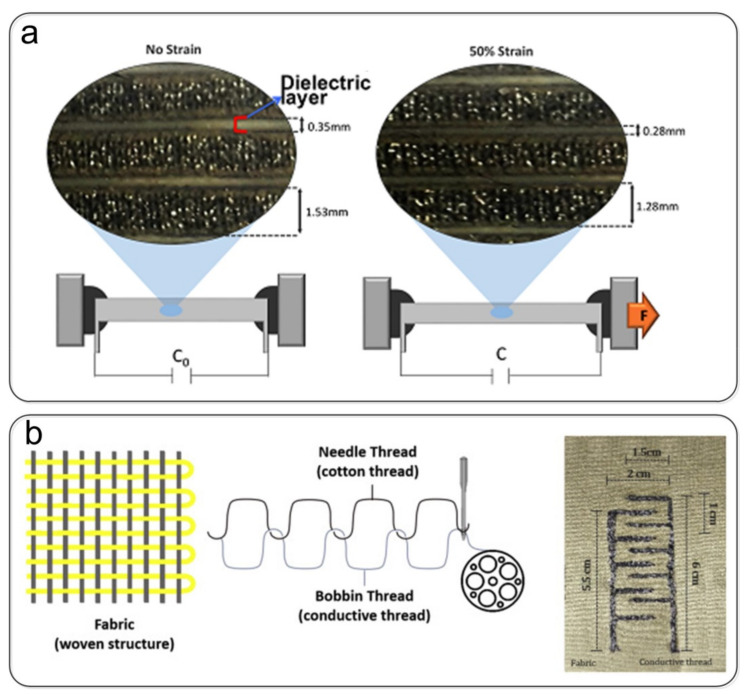
Capacitive pressure sensor based on the in-plane devices. (**a**) Interdigital capacitive strain sensor combining conductive braided fabric and a silicone elastomer. Reproduced with permission from Ref. [137]. Copyright 2021 MDPI. (**b**) Interdigitated capacitor on fabric as a tactile sensor. Reproduced with permission from Ref. [138]. Copyright 2021 Elsevier.

**Figure 10 nanomaterials-12-01495-f010:**
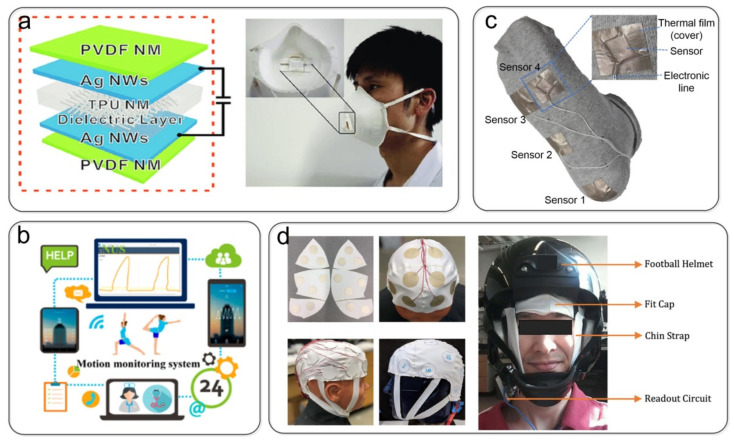
Pressure sensors are used in wearable devices. (**a**) All-textile-structured skin sensors monitor physiological signals, such as human breathing and heart rate. Reproduced with permission from Ref. [106]. Copyright 2017 Wiley-VCH. (**b**) All-textile pressure sensor and wireless batteryless monitoring system for real-time human movement detection. Reproduced with permission from Ref. [107]. Copyright 2019 American Chemical Society. (**c**) Capacitive textile pressure sensor for walking gait analysis. Reproduced with permission from Ref. [108]. Copyright 2020 Elsevier. (**d**) Capacitive pressure sensing array composed of conductive fabric electrodes used to measure and map the pressure on a player’s head wearing a helmet. Reproduced with permission from Ref. [140]. Copyright 2021 American Chemical Society.

**Figure 11 nanomaterials-12-01495-f011:**
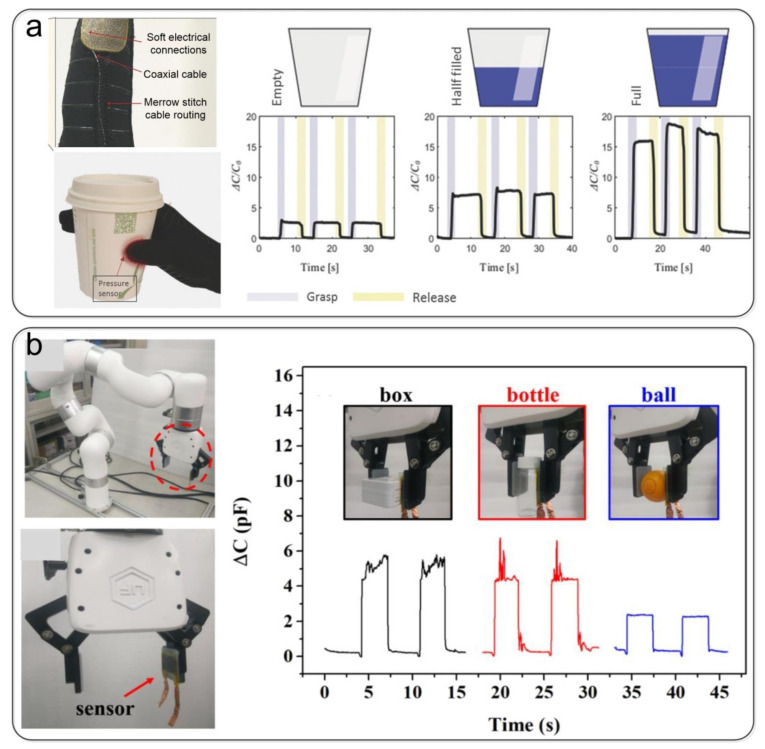
Pressure sensors are used in robotic sensing. (**a**) Textile gloves for grip detection. Reproduced with permission from Ref. [100]. Copyright 2017 Wiley-VCH. (**b**) Real-time monitoring of a robotic arm grasping objects. Reproduced with permission from Ref. [96]. Copyright 2021 Elsevier.

**Figure 12 nanomaterials-12-01495-f012:**
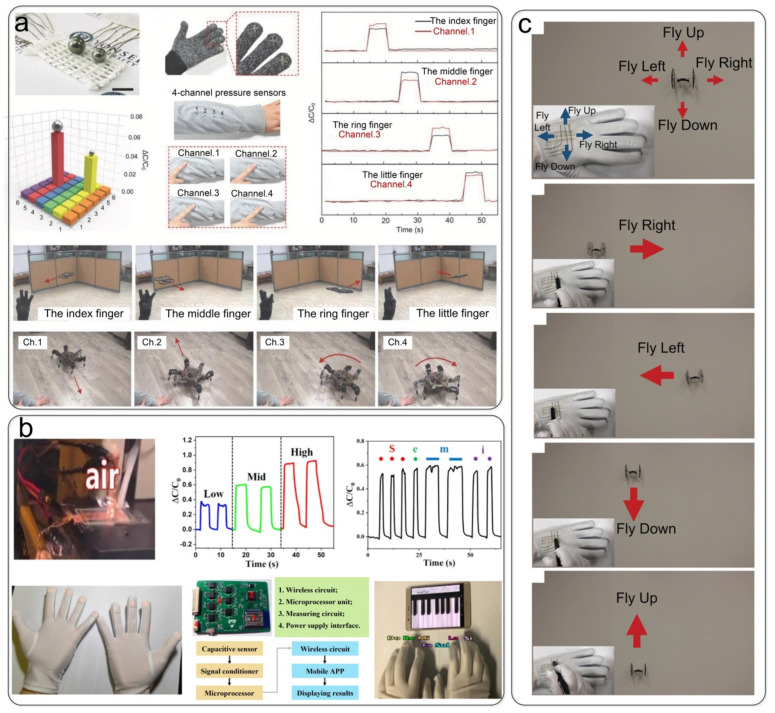
Pressure sensors are used for human–machine interaction. (**a**) The textile capacitive pressure sensor is used for wireless control of a UAV quadrotor and hexapod walking robot. Reproduced with permission from Ref. [111]. Copyright 2015 Wiley-VCH. (**b**) Capacitive tactile sensors are used to play a piano. Reproduced with permission from Ref. [104]. Copyright 2020 American Chemical Society. (**c**) Smart gloves for the remote control of drones. Reproduced with permission from Ref. [130]. Copyright 2020 Wiley-VCH.

**Table 1 nanomaterials-12-01495-t001:** Comparison of the advantages and disadvantages of different types of sensors.

Type	Advantages	Disadvantages	References
Capacitive	Good stability, low power consumption, high response speed, simple structure, and low-cost scalable manufacturing process	Limited sensitivity and easily disturbed by external fields due to the low compressibility of solid media	[58,59]
Resistive	Simple structure and working mechanism, relatively simple manufacturing process, and high response speed	Large signal drift	[33]
Piezoelectric	High sensitivity, fast response, low power consumption, self-powered, dynamic detection, simple structure, and convenient signal acquisition	Some difficulties in the measurement of static force, complicated manufacturing, high cost, and the need for material to be electrically polarized	[15,38]
Triboelectric	Low cost, simple preparation process, high output voltage, simple structure, convenient signal acquisition, and low energy consumption	Unnecessary sensitivity to static electricity, temperature fluctuations, and drift capacitance	[47,48]
Iontronic	The high ionic conductivity, high interfacial capacitance, high sensitivity, and fast response	Easily affected by ambient temperature and humidity	[53,54]

**Table 2 nanomaterials-12-01495-t002:** Performance comparison of capacitive pressure sensors with different textile structures.

Functional Textile Layers	Electrode Layer	Dielectric Layer	Sensitivity	Detection Limit	Response Time	Measuring Range	Reference
Textile-structured electrode	Conductive knit electrodes	Porous silicone elastomer	0.0121 kPa^−1^	0.86 kPa		0–1 MPa	[100]
	Conductive nylon fabric	Ecoflex	0.035 kPa^−1^ (<16 kPa)		0.801 s	0–16 kPa	[101]
	Silver-plated Polyacrylonitrile (PAN) nanofibers	Nylon netting	1.49 kPa^−1^ (<1 kPa)		48 ms	0–10 kPa	[102]
	Conductive cloth tape	Porous PDMS	0.023 kPa^−1^		155 ms	>200 kPa	[10]
	Carbonized cotton fabric (CCF) electrodes	Porous Ecoflex	0.0245 kPa^−1^ (<100 kPa)		0.1 s	0–1 MPa	[103]
Textile-structured dielectric layer	Fe-Zn electrodes	Polylactic-co-glycolic acid and Polycaprolactone membranes	0.863 kPa^−1^ (0–1.86 kPa)	1.24 Pa	251 ms	0–5 kPa	[95]
	PEDOT: PSS/PDMS electrodes	MXene/Poly(vinylidene fluoride-trifluoroethylene) (PVDF-TrFE) nanofibers	0.51 kPa^−1^ (<1 kPa)	1.5 Pa	0.15 s	0–400 kPa	[94]
	Au electrodes	3D AgNW@TPU films	1.21 kPa^−1^ (<5 kPa)	0.9 Pa	100 ms	0–30 kPa	[104]
	Cu tape	Polyimide (PI) nanofiber membranes	2.204 kPa^−1^ (3.5–4.1 Pa)	3.5 Pa	12.5 ms	0–1.388 MPa	[105]
	PDMS microcylinder arrays	Polyvinylidene Fluoride (PVDF) fiber layers	0.60 kPa^−1^ (0–7 kPa)	0.065 Pa	25 ms	0–50 kPa	[96]
All-textile structure	PVDF nanofiber membranes/AgNWs	Thermoplastic polyurethane (TPU) nanofiber membranes	4.2 kPa^−1^ (0–0.4 kPa)	1.6 Pa	26 ms	30 kPa	[106]
	Fabric/Poly(vinyl alcohol) (PVA) template-assisted silver nanofibers (Ag NFs)	3D penetrated fabric	0.108 kPa^−1^ (0–2.5 kPa)			30 kPa	[107]
	Single-walled carbon nanotubes/Silver paste/Spacer fabric	Encapsulation/Polyethylene terephthalate (PET) yarn layers	0.042 kPa^−1^			1000 kPa	[108]
	AgNW/TPU electrospun nanofiber membranes	TPU electrospun nanofiber membranes	7.24 kPa^−1^ (<0.98 kPa)	9.24 Pa	<55 ms	0–50 kPa	[109]
	AgNW/TPU conductive networks	Micropatterned TPU nanofibers	8.31 kPa^−1^ (<1 kPa)	0.5 Pa	27.3 ms	0–80 kPa	[110]
Yarn structure	Poly(styrene-block-butadienstyrene) (SBS)/Ag nanoparticles (AgNP) composite-coated Kevlar fibers	Solid PDMS	0.21 kPa^−1^ (<2 kPa)	8 mg	40 ms	0–3.9 MPa	[111]
	SBS/AgNP composite-coated Twaron fibers	Microporous PDMS	0.278 kPa^−1^ (<2 kPa)	4 mg	340 ms	0–50 kPa	[112]
	Silver fibers	Cotton fibers	0.0397 kPa^−1^ (<0.85 kPa)	3.6 Pa		0–200 kPa	[98]
	Silver fibers	Cotton fibers	8.697 MPa^−1^ (<4.5 kPa)			0–130 kPa	[113]

## Data Availability

Not applicable.

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
