# Peer review of "Textile-Based Flexible Capacitive Pressure Sensors: A Review"

_nanomaterials, 2022, doi:10.3390/nano12091495_

Round 1
Reviewer 1 Report
Authors have highlighted the emerging and core issue, but still there are major issues to be fixed.
Reviews to Authors
- Title must be simple, clearer and nicer.
- Spell out each acronym the first time used in the body of the paper. Spell out acronyms in the Abstract by extending it.
- The abstract can be rewritten to be more meaningful. The authors should add more details about their final results in the abstract. Abstract should clarify what is exactly proposed (the technical contribution) and how the proposed approach is validated.
- What is the motivation of the proposed work?
- Introduction needs to explain the main contributions of the work clearer.
- The novelty of this paper is not clear. The difference between present work and previous Works should be highlighted.
- Authors must explain in detail the introduction section.
- Authors must develop the framework/architecture of the proposed methods
- There is need of flowchart and pseudocode of the proposed techniques
- Proposed methods should be compared with the state-of-the-art existing techniques
- Research gaps, objectives of the proposed work should be clearly justified.
- To improve the Related Work and Introduction sections authors are highly recommended to consider these high-quality research works <AI-Enabled Framework for Fog Computing Driven E-Healthcare Applications. Sensors, 2021, Vol. 21, No. 23, pp. 8039, 2021 >
- English must be revised throughout the manuscript.
- Limitations and Highlights of the proposed methods must be addressed properly
- Experimental results are not convincing, so authors must give more results to justify their proposal.
Author Response
Overall comment: Authors have highlighted the emerging and core issue, but still there are major issues to be fixed.
Overall Reply: We feel great thanks for your professional review work on our article. As you are concerned, we have made extensive corrections to our draft, the details are listed below. Since some of the questions are similar, we have combined our replies.
Comment 1: Title must be simple, clearer and nicer.
Response 1: Thanks for the reviewer’s careful reading. According to the reviewer’s suggestion, we have corrected it.
Title: Textile-Based Flexible Capacitive Pressure Sensors: A Review
- Flexible capacitive pressure sensors (2.1. Working principle, 2.2. Functional textile layers)
3 Materials Classification and fabrication methods for textile layers (3.1 Materials classification, 3.2 Preparation methods (3.2.1 Weaving technology, 3.2.2 Fabric substrate modification, 3.2.3 Electrospinning technology))
4 Textile-based flexible capacitive pressure sensors (4.1 Sandwich devices, 4.2 Yarn devices, 4.3 In-plane devices)
Comment 2: Spell out each acronym the first time used in the body of the paper. Spell out acronyms in the Abstract by extending it.
Response 2: Thanks for the reviewer’s careful reading. According to the reviewer’s suggestion, we have corrected it.
Comment 3: The abstract can be rewritten to be more meaningful. The authors should add more details about their final results in the abstract. Abstract should clarify what is exactly proposed (the technical contribution) and how the proposed approach is validated.
Response 3: Thanks for the reviewer’s valuable suggestion. We have rewritten the Abstract section.
Abstract: Flexible capacitive pressure sensors have been widely used in electronic skin, human movement& health monitoring, and human-machine interactions. Recently, electronic textiles afford a valuable alternative to traditional capacitive pressure sensors due to their merits of flexibility, lightweight, air permeability, low cost, and feasibility to fit various surfaces. The textile-based functional layers can serve as electrodes, dielectrics, and substrates, and different devices with semi-textile or all-textile structures have been well developed. This paper provides a comprehensive review of recent developments in textile-based flexible capacitive pressure sensors. The latest research progresses on textile devices with sandwich structures, yarn structures, and in-plane structures are introduced, and the influences of different device structures on performance are discussed. The applications of textile-based sensors in human wearable devices, robotic sensing, and human-machine interaction are then summarized. Finally, evolutionary trends, future directions, and challenges are highlighted.
Comment 4: What is the motivation of the proposed work?
Response 4:Thanks for the reviewer’s careful reading. The motivation for the proposed work is explained in the article.
Lines 11-15 on Page 2:
Although the research progress of textile-based flexible pressure sensors is dis-cussed in some articles, a comprehensive overview of textile-based flexible ca-pacitive pressure sensors has not been fully reported in any literature. This work aims to fill this knowledge gap, hoping to provide guiding ideas for improving and innovating flexible textile-based capacitive pressure sensors.
Comment 5: Introduction needs to explain the main contributions of the work clearer.
Response 5: According to the reviewer's kind suggestion, we have rewritten the Introduction section to explain the main contributions of our work.
Lines 15-25 on Page 2:
The four key contributions of this paper are presented as follows. First, starting from the working principle of capacitive pressure sensors, the existing textile-based ca-pacitive pressure sensors are divided into five forms according to the different functions of textiles in capacitive pressure sensors. Second, according to the form of the sensor and the selected material, the preparation methods of the textile-based capacitive pressure sensor can be summarized as weaving technology, fabric substrate modification, and electrospinning technology. Third, textile-based capacitive pressure sensors can be divided into the sandwich, yarn, and in-plane devices. Last, the potential applications of textile-based capacitive pressure sensors are summarized from the aspects of human wearable devices, robot sensing, and human–machine interaction.
Comment 6: The novelty of this paper is not clear. The difference between present work and previous Works should be highlighted.
Response 6: Thanks for the reviewer’s kind suggestion. Although the research progress of textile-based flexible pressure sensors is discussed in some articles, a comprehensive overview of textile-based flexible capacitive pressure sensors has not been fully reported in any literature. This work aims to fill this knowledge gap, hoping to provide guiding ideas for improving and innovating textile-based flexible capacitive pressure sensors.
Comment 7: Authors must explain in detail the introduction section.
Response 7: Thanks for the reviewer’s careful reading. According to the reviewer's suggestion, we have rewritten the Introduction section.
Lines 11-32 on Page 2:
Although the research progress of textile-based flexible pressure sensors is dis-cussed in some articles, a comprehensive overview of textile-based flexible ca-pacitive pressure sensors has not been fully reported in any literature. This work aims to fill this knowledge gap, hoping to provide guiding ideas for improving and innovating flexible textile-based capacitive pressure sensors. The four key contributions of this paper are presented as follows. First, starting from the working principle of capacitive pressure sensors, the existing textile-based capacitive pressure sensors are divided into five forms according to the different functions of textiles in capacitive pressure sensors. Second, according to the form of the sensor and the selected material, the preparation methods of the textile-based capacitive pressure sensor can be summarized as weaving technology, fabric substrate modification, and electrospinning technology. Third, textile-based capacitive pressure sensors can be divided into the sandwich, yarn, and in-plane devices. Last, the potential applications of textile-based capacitive pressure sensors are summarized from the aspects of human wearable devices, robot sensing, and human–machine interaction.
This paper reviews the latest developments in capacitive pressure sensors based on textile structures. Section 2 describes the working principle of capacitive pressure sensors and the different forms of textile-based functional layers. In Section 3, materials and fabrication methods are discussed by considering the requirements for flexible electrodes and dielectric layers. Section 4 describes three devices types based on functional textile layers. Section 5 presents practical applications of textile-based capacitive pressure sensors. Finally, in Section 6, conclusions and outlooks are also put forward.
Comment 8: Authors must develop the framework/architecture of the proposed methods.
Comment 9: There is need of flowchart and pseudocode of the proposed techniques.
Responses 8-9: Thanks for the reviewer’s kind suggestion. Our work summarizes different existing methods for fabricating textile-based capacitive pressure sensors and illustrates the advantages and disadvantages of these methods. We didn't develop an entirely new approach and couldn't provide frameworks, flowcharts, and pseudocode, but our work has figures and tables to support the logic and framework of the article.
Comment 10: Proposed methods should be compared with the state-of-the-art existing techniques.
Response 10: Thanks for the reviewer’s suggestion. Our work has summarized the existing common and state-of-the-art methods for fabricating textile-based capacitive pressure sensors and illustrated the advantages and disadvantages of these methods.
Comment 11: Research gaps, objectives of the proposed work should be clearly justified.
Response 11: In the introduction part, due to the lack of summary of the research status of textile-based capacitive pressure sensors, this review aims to comprehensively summarize the device structures, materials, manufacturing methods, and applications of existing textile-based capacitive pressure sensors.
Comment 12: To improve the Related Work and Introduction sections authors are highly recommended to consider these high-quality research works <AI-Enabled Framework for Fog Computing Driven E-Healthcare Applications. Sensors, 2021, Vol. 21, No. 23, pp. 8039, 2021 >.
Response 12: Thanks for the reviewer’s kind suggestion. We read this research carefully and rewritten the Introduction section concerning its relevant content according to the reviewer's suggestion. The research content and main contributions are explained in detail in the introduction section.
Comment 13: English must be revised throughout the manuscript.
Response 13: Thanks for the reviewer’s careful reading. According to the reviewer’s suggestion, we have made corresponding changes.
Comment 14: Limitations and Highlights of the proposed methods must be addressed properly.
Response 14: Thanks for the reviewer’s suggestion. Our work comprehensively summarizes the device structures, materials, fabrication methods, and applications of existing textile-based capacitive pressure sensors, and compares the advantages and disadvantages of different device structures, materials, and fabrication methods. In Conclusions and outlooks, the limitations of textile-based capacitive pressure sensors in fabrication and applications are discussed.
Comment 15: Experimental results are not convincing, so authors must give more results to justify their proposal.
Response 15: Since there is no experimental part of our work, there are no experimental results. However, we have rewritten the Conclusions and outlooks following the reviewer's suggestion.
- Conclusions and outlooks
Textile-based functional layers can introduce both porous airgaps and mi-cro/nanostructures to enhance the performance of flexible capacitive pressure sensors. In addition, textile-based capacitive pressure sensors exhibit excellent flexibility, breathability, and comfort, making them easy to integrate with clothing and have great potential in wearable electronics. This review presents the current research and progresses in textile-based capacitive pressure sensors. According to the sensing principle and device structures, the functional textile layer can be divided into five forms: textile-structured electrodes, textile-structured dielectric layers, all-textile structures, yarn structures, and interdigital electrode structures. Then, materials and fabrication methods for functional textile layers are discussed by considering the requirements for flexible electrodes and dielectric layers, including weaving technology, fabric substrate modification, and electrospinning technology. Three types of devices with the sandwich, yarn, and in-plane structures are discussed for textile-based sensors. Finally, the textile-based capacitive pressure sensor applications in human wearable devices, robot sensing, and human-machine interactions are demonstrated, indicating its great application potential in wearable intelligent electronic devices.
Looking forward to the future, flexible capacitive pressure sensors based on textile structures not only need to achieve excellent sensing performance (sensitivity, response time, repeat stability, etc.) but also need to ensure that their flexibility, air permeability, durability, and other performance meet the requirements of wearable electronic devices. However, there are still many problems to be solved in the device fabrication process of the existing devices:
- The fabrication method needs to be further optimized. With the development of interdisciplinary integration, electrospinning has become a simple and effective method of preparing pressure sensors with high performance. However, some problems still need to be studied in-depth to apply electrospun nanofiber membranes in the wearable electronics field. For example, how can conductive nanofiber mem-brane-based electrodes be efficiently prepared without affecting their structure, and at the same time, how can repeatability and durability be ensured? How to further improve the compressibility and stretchability of dielectrics is another concern.
- Regarding applications, textile-based capacitive pressure sensors have many applications in intelligent textile apparel. However, the low-cost and efficient prepara-tion of large-area sensor arrays, the reduction in crosstalk between capacitive signals, the improvement in the spatial resolution of the sensor, and multimodal detection still need further research.
3. It is necessary to broaden the exploration of new materials and develop new processes to improve the performance of sensors in the future. Moreover, multifunctional integrated research can be carried out on textile-based sensors to develop comfortable, low-cost smart textile apparel with multiple functions.

Reviewer 2 Report
Review report on “Progress in Textile-Based Flexible Capacitive Pressure Sensors: A Review”
by Min Su at al
The paper is focused on the recent advances of capacitive pressure sensors based on textile structures. Different forms of capacitive pressure sensors are reviewed with detailed explanation of their working principle followed by the material selection and preparation methods. In addition, flexible textile capacitive pressure sensor applications are presented, including human wearable devices and human computer interactions. In the end of the review, the challenges and prospects of flexible textile capacitive pressure sensors are discussed.
The review is well written and presented information is compact and summarized. At the same time, the reviewer is written more like story, where scientific discussion and challenges are missing
In addition, I have the following comments and questions
- Abstract need to be rewritten (looks like Introduction)
- In Table 1. “Comparison of the advantages and disadvantages of different types of sensors” -need to point out which type of sensors?
- May the authors specify their own contribution (as experts) in the field if any?
Author Response
Overall Comment: The paper is focused on the recent advances of capacitive pressure sensors based on textile structures. Different forms of capacitive pressure sensors are reviewed with detailed explanation of their working principle followed by the material selection and preparation methods. In addition, flexible textile capacitive pressure sensor applications are presented, including human wearable devices and human computer interactions. In the end of the review, the challenges and prospects of flexible textile capacitive pressure sensors are discussed. The review is well written and presented information is compact and summarized. At the same time, the reviewer is written more like story, where scientific discussion and challenges are missing.
Overall Reply: We feel great thanks for your professional review work on our article. As you are concerned, we have made extensive corrections to our draft, and the details are listed below.
Comment 1: Abstract need to be rewritten (looks like Introduction).
Response 1: Thanks for the reviewer’s careful reading. Considering the reviewer's kind suggestion, we have corrected it.
Abstract: Flexible capacitive pressure sensors have been widely used in electronic skin, human movement& health monitoring, and human-machine interactions. Recently, electronic textiles afford a valuable alternative to traditional capacitive pressure sensors due to their merits of flexibility, lightweight, air permeability, low cost, and feasibility to fit various surfaces. The textile-based functional layers can serve as electrodes, dielectrics, and substrates, and different devices with semi-textile or all-textile structures have been well developed. This paper provides a comprehensive review of recent developments in textile-based flexible capacitive pressure sensors. The latest research progresses on textile devices with sandwich structures, yarn structures, and in-plane structures are introduced, and the influences of different device structures on performance are discussed. The applications of textile-based sensors in human wearable devices, robotic sensing, and human-machine interaction are then summarized. Finally, evolutionary trends, future directions, and challenges are highlighted.
Comment 2: In Table 1. “Comparison of the advantages and disadvantages of different types of sensors” -need to point out which type of sensors?
Response 2: Thanks for the reviewer’s careful reading. The types of sensors are divided into capacitive, resistive, piezoelectric, triboelectric, and iontronic. Considering the reviewer's suggestion, we describe the type of sensor in the article. The corresponding types are also mentioned in Table 1.
Page 3:
2.1. Working principle
Table 1 compares the advantages and disadvantages of different types of sensors, including capacitive, resistive, piezoelectric, triboelectric, and iontronic.
Table 1. Comparison of the advantages and disadvantages of different types of sensors.
Type |
Advantages |
Disadvantages |
References |
Capacitive |
Good stability, low power consumption, high response speed, simple structure, and low-cost scalable manufacturing process |
Limited sensitivity and easily disturbed by external fields due to the low compressibility of solid media |
[53, 54] |
Resistive |
Simple structure and working mechanism, relatively simple manufacturing process, and high response speed |
Large signal drift |
[31] |
Piezoelectric |
High sensitivity, fast response, low power consumption, self-powered, dynamic detection, simple structure, and convenient signal acquisition |
Some difficulties in the measurement of static force, complicated manufacturing, high cost, and the need for material to be electrically polarized |
[15, 36] |
Triboelectric |
Low cost, simple preparation process, high output voltage, simple structure, convenient signal acquisition, and low energy consumption |
Unnecessary sensitivity to static electricity, temperature fluctuations, and drift capacitance |
[45,46] |
Iontronic |
The high ionic conductivity, high interfacial capacitance, high sensitivity, and fast response |
Easily affected by ambient temperature and humidity |
[51, 52] |
Comment 3: May the authors specify their own contribution (as experts) in the field if any?
Response 3: Thanks for the reviewer’s kind suggestion. We have unique insights into fabricating high-quality flexible sensors and special applications from material preparation to micro-nano structure design. We successfully fabricated thin films with microstructures and applied them to various flexible sensing devices. Such as pressure sensor (ACS Applied Materials & Interfaces, 2021, 13, 20448; Nano Energy 2020, 105580; ACS Applied Materials & Interfaces 2019, 11,14997; ACS Applied Materials & Interfaces, 2016, 8, 16869; Nanomaterials 2019, 9,496), strain sensor (Journal of Materials Chemistry C 2015, 3, 12379; Nanotechnology 2017, 28, 115501), temperature sensor (Nanophotonics 2018, 7, 883-892; Rsc Advances 2015, 5, 25609), humidity sensors (Nanoscale 2017,9, 6246), biosensors (Sensors and Actuators B: Chemical 2017, 250, 333; Biomaterials 2017, 133, 49; Nanotechnology 2017, 28, 315501), etc. Besides, we have developed some meaningful researches on textile electronics, including large-scale washable smart textiles based on triboelectric nanogenerator arrays for self‐powered sleeping monitoring (Advanced Functional Materials, 2018, 28, 1704112), flexible pressure sensors base on stretchable conductive textiles (Journal of Physics D: Applied Physics, 2022), and all-textile capacitive pressure sensors based on piezoelectric nanofibers for wearable electronics and robotic sensing.

Round 2
Reviewer 1 Report
Authors have improved the paper at some extent but still there are major comments to be fixed
Major Comments
- Introduction needs to explain the main contributions of the work clearer.
- The novelty of this paper is not clear. The difference between present work and previous Works should be highlighted.
- Authors must explain in detail the introduction section.
- Authors must develop the framework/architecture of the proposed methods
- There is need of flowchart and pseudocode of the proposed techniques
- Proposed methods should be compared with the state-of-the-art existing techniques
- Research gaps, objectives of the proposed work should be clearly justified.
- To improve the Related Work and Introduction sections authors are highly recommended to consider these high quality research works <AI-driven adaptive reliable and sustainable approach for Internet of Things enabled healthcare system’ >
- English must be revised throughout the manuscript.
- Limitations and Highlights of the proposed methods must be addressed properly
- Experimental results are not convincing, so authors must give more results to justify their proposal.
Author Response
Overall comment: Authors have improved the paper at some extent but still there are major comments to be fixed
Overall Reply: We feel great thanks for your kind comments. Our article is a comprehensive review for textile-based flexible capacitive pressure sensors, which cannot provide framework, pseudocode, and experimental results. Nonetheless, we have carefully answered your questions point-to-point in Round 1 and Round 2. Thus, we expect you to carefully evaluate our replies. As you are concerned, we have made extensive corrections to our draft, the details are listed below.
Comment 1: Introduction needs to explain the main contributions of the work clearer.
Response 1: Thanks for the reviewer's kind suggestion. According to the reviewer's kind suggestion, the main contributions of our work are presented in the Introduction.
Lines 15-25 on Page 2:
The four key contributions of this paper are presented as follows. First, starting from the working principle of capacitive pressure sensors, the existing textile-based capacitive pressure sensors are divided into five forms according to the different functions of textiles in capacitive pressure sensors. Second, according to the form of the sensor and the selected material, the preparation methods of the textile-based capacitive pressure sensor can be summarized as weaving technology, fabric substrate modification, and electrospinning technology. Third, textile-based capacitive pressure sensors can be divided into sandwich, yarn, and in-plane devices. Last, the potential applications of textile-based capacitive pressure sensors are summarized from the aspects of human wearable devices, robot sensing, and human–machine interaction.
Comment 2: The novelty of this paper is not clear. The difference between present work and previous Works should be highlighted.
Response 2: Thanks for the reviewer's kind suggestion. Our paper is a review article, and the novelty and purpose of this paper are stated in the article.
Lines 11-15 on Page 2:
Although the research progress of textile-based flexible pressure sensors is discussed in some articles, a comprehensive overview of textile-based flexible capacitive pressure sensors has not been fully reported in the literature. This work aims to fill this knowledge gap, hoping to provide guiding ideas for improving and innovating flexible textile-based capacitive pressure sensors.
Comment 3: Authors must explain in detail the introduction section.
Response 3: Thanks for the reviewer's careful reading. According to the reviewer's suggestion, we explain the main contributions and content of the work in the Introduction.
Lines 11-32 on Page 2:
The four key contributions of this paper are presented as follows. First, starting from the working principle of capacitive pressure sensors, the existing textile-based capacitive pressure sensors are divided into five forms according to the different functions of textiles in capacitive pressure sensors. Second, according to the form of the sensor and the selected material, the preparation methods of the textile-based capacitive pressure sensor can be summarized as weaving technology, fabric substrate modification, and electrospinning technology. Third, textile-based capacitive pressure sensors can be divided into sandwich, yarn, and in-plane devices. Last, the potential applications of textile-based capacitive pressure sensors are summarized from the aspects of human wearable devices, robot sensing, and human–machine interaction.
This paper reviews the latest developments in capacitive pressure sensors based on textile structures. Section 2 describes the working principle of capacitive pressure sensors and the different forms of textile-based functional layers. In Section 3, materials and fabrication methods are discussed by considering the requirements for flexible electrodes and dielectric layers. Section 4 describes three devices types based on functional textile layers. Section 5 presents practical applications of textile-based capacitive pressure sensors. Finally, in Section 6, conclusions and outlooks are also put forward.
Comment 4: Authors must develop the framework/architecture of the proposed methods.
Response 4: Thanks for the reviewer's kind suggestion. The proposed method is for the fabrication of sensors, so the framework/architecture of the proposed method cannot be developed. Here we provide the framework for the article.
Lines 81 on Page 2:
Figure 1 shows the framework of the textile-based capacitive pressure sensor.
Figure 1 Framework of textile-based capacitive pressure sensors
Comment 5: There is need of flowchart and pseudocode of the proposed techniques.
Response 5: Thanks for the reviewer's kind suggestion. The proposed techniques and methods are for efficiently fabricating flexible sensors; technical flowcharts and pseudocodes cannot be provided. The corresponding preparation process has been introduced in the article. For example, Figure 6 shows the preparation process of electrospinning technology.
Figure 6. Capacitive pressure sensor prepared by electrospinning technology. (a) Screen printing technology coated CNTs on Electrospun TPU nanofiber membranes as planar electrodes. Reproduced with permission from ref[104]. Copyright 2021, MDPI. (b) Electrospinning with Pd2+/PAN solution and electroless plating of a mixed nanofiber membrane to prepare a flexible electrode. Reproduced with permission from ref[97]. Copyright 2020, Elsevier. (c) Preparation of a dual-structure polyurethane nanofiber membrane by electrospinning. Reproduced with permission from ref[129]. Copyright 2022, American Chemical Society. (d) Preparation of an MXene/PVDF-TrFE composite nanofiber membrane and its use as the dielectric layer of a capacitive pressure sensor. Reproduced with permission from ref88. Copyright 2020, American Chemical Society.
Comment 6: Proposed methods should be compared with the state-of-the-art existing techniques.
Response 6: Thanks for the reviewer's suggestion. Our work has summarized the existing common and state-of-the-art methods for fabricating textile-based capacitive pressure sensors and illustrated the advantages and disadvantages of these methods. For example, the performance of devices is compared in Table 2. The advantages and disadvantages of different preparation methods are also discussed in the article.
Lines 11-25 on Page 12:
In general, these studies suggest that in the preparation of flexible textile capacitive pressure sensors, knitting retains the original textile structure of the clothing, making the fabricated sensor more suitable for three-dimensional curved surfaces and more flexibly integrated into wearable electronic products. However, the structural design and performance optimization of the fibers or yarns used for weaving needs to be further studied, and the problem of sensor slippage under pressure has yet to be resolved. The method of fabric substrate modification is relatively simple, and the selection of conductive materials and attachment methods is relatively flexible. However, the uneven coating and easy shedding between the fabric substrate and the conductive material damages the fabric's original textile structure and affects its conductivity. Electrospinning can control the thickness and porosity of the fiber membrane through parameter settings. The fiber membrane and the coating material can be better combined through material selection, and the sensor performance and use performance can be further improved.
Comment 7: Research gaps, objectives of the proposed work should be clearly justified.
Response 7: Thanks for the reviewer's kind suggestion. The objectives of the proposed work are presented in the Introduction.
Lines 11-15 on Page 2:
Although the research progress of textile-based flexible pressure sensors is discussed in some articles, a comprehensive overview of textile-based flexible capacitive pressure sensors has not been fully reported in the literature. This work aims to fill this knowledge gap, hoping to provide guiding ideas for improving and innovating flexible textile-based capacitive pressure sensors.
Comment 8: To improve the Related Work and Introduction sections authors are highly recommended to consider these high quality research works <AI-driven adaptive reliable and sustainable approach for Internet of Things enabled healthcare system'>.
Response 8: Thanks for the reviewer's kind suggestion. We have read these high quality research works carefully and revised the Introduction, Conclusions, etc. And we have cited them in our article.
- Sodhro, A. H.; Zahid, N., AI-Enabled Framework for Fog Computing Driven E-Healthcare Applications. Sensors 2021, 21, 8039.
- Zahid, N.; Sodhro, A. H.; Kamboh, U. R.; Alkhayyat, A.; Wang, L., AI-driven adaptive reliable and sustainable approach for internet of things enabled healthcare system. Math Biosci Eng 2022, 19, 3953-3971.
Comment 9: English must be revised throughout the manuscript.
Response 9: Thanks for the reviewer's careful reading. According to the reviewer's suggestion, we have made corresponding changes.
Comment 10: Limitations and Highlights of the proposed methods must be addressed properly.
Response 10: Thanks for the reviewer's suggestion. Our work compares the advantages and disadvantages of different device structures, materials, and fabrication methods. Different methods will be selected to prepare different devices according to the actual application scenarios. Highlights of the article are presented in the Introduction section. In Conclusions and outlooks, the limitations of textile-based capacitive pressure sensors in fabrication and applications are discussed.
Lines 15-25 on Page 2:
The four key contributions of this paper are presented as follows. First, starting from the working principle of capacitive pressure sensors, the existing textile-based capacitive pressure sensors are divided into five forms according to the different functions of textiles in capacitive pressure sensors. Second, according to the form of the sensor and the selected material, the preparation methods of the textile-based capacitive pressure sensor can be summarized as weaving technology, fabric substrate modification, and electrospinning technology. Third, textile-based capacitive pressure sensors can be divided into sandwich, yarn, and in-plane devices. Last, the potential applications of textile-based capacitive pressure sensors are summarized from the aspects of human wearable devices, robot sensing, and human–machine interaction.
Lines 5-27 on Page 20:
Textile-based flexible capacitive pressure sensors need to achieve excellent sensing performance (sensitivity, response time, repeat stability, etc.) and ensure that their flexibility, air permeability, durability, and other performance meet the requirements of wearable electronic devices. However, there are still many problems to be solved in the device fabrication process of the existing devices:
- The fabrication method needs to be further optimized. With the development of interdisciplinary integration, electrospinning has become a simple and effective method of preparing pressure sensors with high performance. However, some problems still need to be studied in-depth to apply electrospun nanofiber membranes in the wearable electronics field. For example, how can conductive nanofiber mem-brane-based electrodes be efficiently prepared without affecting their structure, and at the same time, how can repeatability and durability be ensured? How to further improve the compressibility and stretchability of dielectrics is another concern.
- Regarding applications, textile-based capacitive pressure sensors have many applications in intelligent textile apparel. However, the low-cost and efficient prepara-tion of large-area sensor arrays, the reduction in crosstalk between capacitive signals, the improvement in the spatial resolution of the sensor, and multimodal detection still need further research.
- It is necessary to broaden the exploration of new materials and develop new processes to improve the performance of sensors in the future. Moreover, multifunctional integrated research can be carried out on textile-based sensors to develop comfortable, low-cost smart textile apparel with multiple functions.
Comment 11: Experimental results are not convincing, so authors must give more results to justify their proposal.
Response 11: Thanks for the reviewer's kind suggestion. Our review article does not cover the experimental part, so there are no experimental results. Our work compares the advantages and disadvantages of different device structures, materials, and fabrication methods. There are figures and tables in the article to support our conclusions, such as Table 2.
Table 2. Performance comparison of capacitive pressure sensors with different textile structures
Functional textile layers |
Electrode layer |
Dielectric layer |
Sensitivity |
Detection limit |
Response time |
Measuring range |
Reference |
Textile-structured electrode
|
Conductive knit electrodes |
Porous silicone elastomer |
0.0121 kPa-1 |
0.86 kPa |
|
0–1 MPa |
[97] |
Conductive nylon fabric |
Ecoflex |
0.035 kPa-1 (<16 kPa) |
|
0.801 s |
0–16 kPa |
[98] |
|
Silver-plated Polyacrylonitrile (PAN) nanofibers |
Nylon netting |
1.49 kPa-1 (<1 kPa) |
|
48 ms |
0–10 kPa |
[99] |
|
Conductive cloth tape |
Porous PDMS |
0.023 kPa-1 |
|
155 ms |
> 200 kPa |
[10] |
|
Carbonized cotton fabric (CCF) electrodes |
Porous Ecoflex |
0.0245 kPa-1 (<100 kPa) |
|
0.1 s |
0–1 MPa |
[100] |
|
Textile-structured dielectric layer |
Fe-Zn electrodes |
Polylactic-co-glycolic acid and Polycaprolactone membranes |
0.863 kPa-1 (0–1.86 kPa) |
1.24 Pa |
251 ms |
0–5 kPa |
[92] |
PEDOT:PSS/PDMS electrodes |
MXene/Poly(vinylidene fluoride-trifluoroethylene) (PVDF-TrFE) nanofibers |
0.51 kPa-1 (<1 kPa) |
1.5 Pa |
0.15 s |
0–400 kPa |
[91] |
|
Au electrodes |
3D AgNW@TPU films |
1.21 kPa-1 (<5 kPa) |
0.9 Pa |
100 ms |
0–30 kPa |
[101] |
|
Cu tape |
Polyimide(PI) nanofiber membranes |
2.204 kPa-1 (3.5–4.1 Pa) |
3.5 Pa |
12.5 ms |
0–1.388 MPa |
[102] |
|
PDMS microcylinder arrays |
Polyvinylidene Fluoride(PVDF) fiber layers |
0.60 kPa-1 (0–7 kPa) |
0.065 Pa |
25 ms |
0–50 kPa |
[93] |
|
All-textile structure |
PVDF nanofiber membranes / AgNWs |
Thermoplastic polyurethane(TPU) nanofiber membranes |
4.2 kPa-1 (0–0.4 kPa) |
1.6 Pa |
26 ms |
30 kPa |
[103] |
Fabric/Poly(vinyl alcohol) (PVA) template-assisted silver nanofibers (Ag NFs) |
3D penetrated fabric |
0.108 kPa-1 (0–2.5 kPa) |
|
|
30 kPa |
[104] |
|
Single-walled carbon nanotubes/Silver paste/Spacer fabric |
Encapsulation/Polyethylene terephthalate(PET) yarn layers |
0.042 kPa-1 |
|
|
1000 kPa |
[105] |
|
AgNW/TPU electrospun nanofiber membranes |
TPU electrospun nanofiber membranes |
7.24 kPa-1 (<0.98 kPa) |
9.24 Pa |
<55 ms |
0–50 kPa |
[106] |
|
AgNW/TPU conductive networks |
Micropatterned TPU nanofibers |
8.31 kPa-1 (<1 kPa) |
0.5 Pa |
27.3 ms |
0–80 kPa |
[107] |
|
Yarn structure |
Poly(styrene-block-butadienstyrene) (SBS)/Ag nanoparticles (AgNP) composite-coated Kevlar fibers |
Solid PDMS |
0.21 kPa-1 (< 2 kPa) |
8 mg |
40 ms |
0–3.9 MPa |
[108] |
SBS/AgNP composite-coated Twaron fibers |
Microporous PDMS |
0.278 kPa-1 (< 2 kPa) |
4 mg |
340 ms |
0–50 kPa |
[109] |
|
Silver fibers |
Cotton fibers |
0.0397 kPa-1 (< 0.85 kPa) |
3.6 Pa |
|
0–200 kPa |
[95] |
|
Silver fibers |
Cotton fibers |
8.697 MPa-1 (<4.5 kPa) |
|
|
0–130 kPa |
[110] |

Round 3
Reviewer 1 Report
Authors have improved the paper, so minor changes are required
- Fig 1 can be redrawn with high visibility, and more details in caption
- Limitations and future recommendations of the conducted survey must be highlighted
Author Response
Overall comment: Authors have improved the paper, so minor changes are required.
Overall Reply: We feel great thanks for your professional review work on our article. As you are concerned, we have made changes to the manuscript, and details are as follows.
Comment 1: Fig 1 can be redrawn with high visibility, and more details in caption.
Response 1: Thanks for the reviewer's kind suggestion. According to the reviewer's suggestion, we have revised Figure 1.
Figure 1. The framework of textile-based capacitive pressure sensors.
Comment 2: Limitations and future recommendations of the conducted survey must be highlighted.
Response 2: According to your kind suggestion, limitations and recommendations for the future are highlighted in the Conclusions and Outlook section.
Lines 8-27 on Page 20:
However, there are still many limitations in the fabrication process and application of the existing devices:
- The fabrication method needs to be further optimized. Problems such as easy peeling and unevenness of conductive materials on the surface of the textile-based electrode will affect their conductivity and durability. Therefore, there is a need to develop an efficient method for fabricating textile-based electrodes. In addition, the dielectric properties, compressibility, and stretchability of textile-based dielectric layers also need to be further improved.
- Although textile-based capacitive pressure sensors have many applications in smart textile clothing, current smart clothing is not washable. Furthermore, the low-cost and efficient fabrication of large-area textile-based capacitive pressure sensor arrays has not been reported yet. Other aspects, such as reducing crosstalk between capacitive signals, multimodal detection, etc., still need further research.
In the future, the way to improve the performance of sensors lies in broadening the exploration of new materials and developing new processes. Moreover, multifunctional integration studies of textile-based sensors can be conducted for the development of comfortable, low-cost multifunctional smart textile garments.
